# Dynamically Triggering Resilient Control for Networked Nonlinear Systems under Malicious Aperiodic DoS Attacks

**Wei Tan [1,*], He Wang [1], Huazhou Hou [1], Xiaoxu Liu [2] and Meng Zheng [3]**

[1]  School of Mathematics, Southeast University, Nanjing 211189, China
[2]  Sino-German College of Intelligent Manufacturing, Shenzhen Technology University, Shenzhen 518118, China;
[3]  Shenyang Institute of Automation, Chinese Academy of Sciences, Shenyang 110016, China
[*]  Correspondence: tanweilina@163.com

**Abstract:** Networked nonlinear systems (NNSs) have great potential security threats because of malicious attacks. These attacks will destabilize the networked systems and disrupt the communication to the networked systems, which will affect the stability and performance of the networked control systems. Therefore, this paper aims to deal with the resilient control problem for NNSs with dynamically triggering mechanisms (DTMs) and malicious aperiodic denial-of-service (DoS) attacks. To mitigate the impact from DoS attacks and economize communication resources, a resilient dynamically triggering controller (RDTC) is designed with DTMs evolving an adaptive adjustment auxiliary variable. Thus, the resulting closed-loop system is exponentially stable by employing the piecewise Lyapunov function technique. In addition, according to the minimum inter-event time, the Zeno behavior can be excluded. Finally, the merits of the proposed controllers and theory are corroborated using the well-known nonlinear Chua circuit.

**Keywords:** resilient dynamically triggering controller (DTRC); dynamically triggering mechanisms (DTMs); denial-of-service (DoS); networked nonlinear systems (NNSs); resilient dynamically event-triggering (RDET)





## 1. Introduction

Recently, due to the irreplaceable position of communication in the network, many scholars have devoted their attention to the study of networked control systems. More specifically, they are focusing on the data transmission of networked control systems because of the advantages of the information interaction based on the interconnection of the different systems. Naturally, communication-based issues of control and optimization are emerging and developing rapidly [1–3]. Although the advantages of communication technology have brought seismic shocks to academia and industry, there exists a problem. The limitations of the periodic sampling technique subject to guaranteeing desired system performance by reducing the sampling period create a large amount of redundant sampled data, resulting in network congestion and executing control tasks periodically after the system is stabilized, which results in wasting network bandwidth and computation resources. To tackle these limitations, an event-triggering technique has emerged at this historical moment and developed rapidly based on supervising the controller's update [4,5]. Herein, it is worth noting that the event triggering techniques not only ensure the desired performance from control tasks but also decrease the update frequency of the controller, resulting in energy saving in system communication. In the past decades, different types of the event triggering techniques have been proposed, such as the static event-triggering technique [6–8], dynamic event-triggering technique [9–11], stochastic event-triggering technique [12–14], and switched event-triggering technique [15,16].

In addition, unreliable communication channels cause much concern in the discussion of the stability and performance maintenance for networked control systems. In this regard,

there exist some innovative works [17–19]. In particular, security-based networked control systems resisting malicious attacks have been given attention in the past years [20,21]. Herein, it is worth noting that on the premise of ensuring system stability and desired performance, the so-called security is the elasticity of resisting malicious cyber attacks. Although there exist a few innovative and groundbreaking results for networked control systems with co-design of the event triggering techniques and DoS attacks [22–24], the co-design with dynamically triggering techniques and DoS attacks for NNSs is still a challenging problem.

Up to now, although many innovative and groundbreaking results have sprung up for the control and optimization in the framework of co-design of dynamically triggering techniques and DoS attacks [25–27], they primarily focus on linear systems. Moreover, some research results of nonlinear systems with DTMs and DoS attacks appear sporadically [28–31], the existing DTMs have shown certain limitations in theorem research and industrial practice. Inspired by the aforementioned discussion, this paper deals with the resilient control problem for NNSs with DTMs and DoS attacks. More specifically, a DTRC is designed with DTMs with an adaptive adjustment auxiliary variable, which can result in the closed-loop system being exponentially stable by employing the piecewise Lyapunov function technique. Meanwhile, a minimal inter-event time is obtained to ensure it is Zeno-free under aperiodic DoS attacks. In addition, the innovations of this paper are as follows:

- Different from the static trigger strategy in [22–24], a novel dynamically triggering strategy is proposed for NNSs with aperiodic DoS attacks. Because of the longer trigger intervals compared with the static trigger intervals, this strategy further reduces the sampling data transmission rate and improves the usage of network resources.
- Compared with the trigger strategy in [23], the dynamically triggering resilient control strategy is introduced into nonlinear systems to obtund the influence of aperiodic DoS attacks. In addition, the sampled data cannot be transmitted even if condition (7) is satisfied since aperiodic DoS attacks will result in the loss of control input during the DoS attacks range.
- Compared with [23], a new piecewise Lyapunov function is designed to ensure the exponential stability of the networked control system under DoS attacks. In particular, the minimum inter-event time excludes Zeno behavior in the resilient controller. Moreover, the proposed method not only releases Assumption 4 in [23] but also reduces the conservative of the system.

The structure of this article is as follows. First, NNSs and problem statements are presented in Section 2. Then, the conditions for the stability of NNSs under DoS attacks are driven in Sections 3. Furthermore, the satisfactory and better performance of the RDET controller designed than the existing ones is provided in Section 4. Finally, the conclusion is shown in Section 5.

*Notations*: $\mathbb{R}^+$ and $\mathbb{Z}^+$ represent the set of the positive real numbers and the set of the positive integer numbers, respectively. $\mathbb{R}^n$ and $\mathbb{R}^{n \times m}$ indicate the space of real $n$-vectors and $n \times m$ matrices. $x^{-1}$ is the inverse of $x$ (function or matrix). $\| \cdot \|$ means the 2-norm.

## 2. Problem Formulation

Figure 1 shows wireless NNSs under aperiodic DoS attacks. First, aperiodic DoS attack scenarios are typically depicted by the sleeping intervals and DoS attack intervals in Figure 2. Then, the dynamically triggering resilient control strategy and switching controller are designed for NNSs, respectively. Next, based on these descriptions, we give out the problem statement.

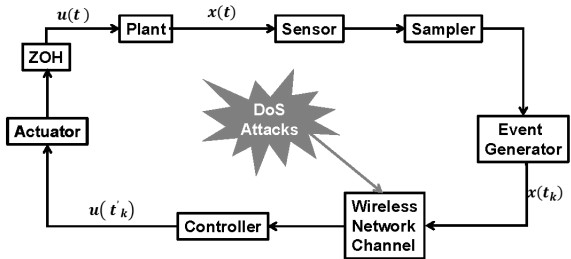

**Figure 1.** Schematic representation of dynamic event-triggered control for nonlinear systems under DoS attacks.

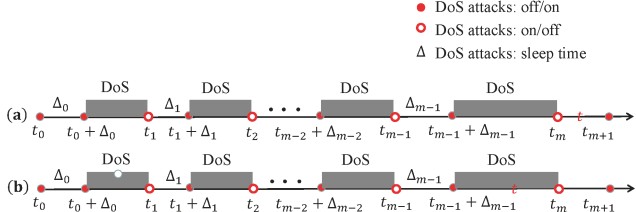

**Figure 2.** Situation of aperiodic DoS attacks. (**a**) presents the current instant in the sleeping interval, and (**b**) presents the current instant in the DoS attack interval.

### 2.1. Networked Nonlinear Systems

Consider the following class of NNSs

$$
\begin{cases}
\dot{x}(t) = f(x(t), u(t)) \\
x(0) = x_0
\end{cases}
\tag{1}
$$

where $x(t) \in \mathbb{R}^n$ is the system state; $u(t) \in \mathbb{R}^m$ represents the control input; $x_0$ denotes the initial condition; the Lipschitz continuous function $f : \mathbb{R}^n \times \mathbb{R}^m \to \mathbb{R}^n$ satisfies $f(0,0) = 0$ for all $t \in \mathbb{R}^+$.

For the sake of later analysis, we provide the following definitions and assumptions.

**Definition 1** ([32])**.**

1.  *A function $\alpha : [0, \infty) \to [0, \infty)$ is called a class of $\mathcal{K}$ if it is continuous, strictly increasing and $\alpha(0) = 0$. If $\alpha \in \mathcal{K}$ and also $\alpha(s) \to [0, \infty)$ as $s \to [0, \infty)$, then it is said to be of class $\mathcal{K}_\infty$.*
2.  *A function $\beta : [0, \infty) \times [0, \infty) \to [0, \infty)$ is called a class of $\mathcal{KL}$ if the function $\beta(\cdot, t) \in \mathcal{K}$ for all fixed $t > 0$ and the function $\beta(s, \cdot)$ is decreasing and $\beta(s, t) \to 0$ as $t \to \infty$ for all fixed $s \in \mathbb{R}^+$.*

**Definition 2** ([32])**.** *System (1) is said to be globally weakly exponentially stable (GWES) if there exist functions $\alpha_1, \alpha_2 \in \mathcal{K}_\infty$ and constants $\delta > 0, M \geq 1$ such that for any initial value $x(0)$, the solution $x(t)$ satisfies $\alpha_1(\|x(t)\|) \leq M e^{-\delta t} \alpha_2(\|x(0)\|), \forall t \geq 0$. In particular, when $\alpha_1(\|x\|) = \alpha_2(\|x\|) = \|x\|^m, m \in \mathbb{Z}^+$, it is said to be globally exponentially stable (GES).*

**Definition 3** ([32])**.** *A function $V : \mathbb{R}^n \to \mathbb{R}_{\geq 0}$ is an ISS-Lyapunov function if there exist some $\mathcal{K}_\infty$ functions $\alpha_1$, $\alpha_2$, and $\gamma$ that satisfy*

$$
\alpha_1(\|x\|) \leq V(x) \leq \alpha_2(\|x\|)
$$
$$
\nabla V(x) f(x, u) \leq -cV(x) + \gamma(\|u\|)
$$

*where $c \in \mathbb{R}^+$.*

**Definition 4** ([32]). *Given a local Lipschitz function $V : \mathbb{R}^n \to \mathbb{R}^+$, the upper left-hand Dini derivative of V along system* (1) *is defined by*

$$D^- V[f] = \lim_{h \to 0^-} \sup \frac{1}{h} \{V(x + hf) - V(x)\}.$$

*2.2. Aperiodic DoS Attacks*

In this section, for the convenience of analysis and design, we assume that the DoS attacks only occur on the measurement channel and that no packet loss or no delay occurs during the sleeping intervals. As shown in Figure 2, $\{t_m\}(t_0 \geq 0)$ and $\{\Delta_m\}_{m \in \mathbb{N}_0}(\Delta_m \geq 0)$ represent the sequence of DoS on/off transitions times and the duration of the $m$th sleep status, respectively. Correspondingly, for simplicity, let $\mathcal{H}_m$ and $\mathcal{D}_m$ present the sleeping interval and the attack interval, respectively.

$$\begin{cases} \mathcal{H}_m = [t_m, t_m + \Delta_m) \\ \mathcal{D}_m = [t_m + \Delta_m, t_{m+1}) \end{cases} \tag{2}$$

In addition, let $\Xi_s(0, t)$ and $\Xi_a(0, t)$ represent, respectively, all single sleeping ranges and all single attack ranges

$$\begin{cases} \Xi_s(0, t) = \{t_m\} \cup \mathcal{H}_m \cap [0, t) \\ \Xi_a(0, t) = \cup \mathcal{D}_m \cap [0, t). \end{cases} \tag{3}$$

To characterize the constraints of limited energy on DoS attacks, it is necessary to give the following two assumptions for the frequency and the duration of DoS attacks, respectively.

**Assumption 1** (*DoS Duration* [22]). *There exist $T_0 \in \mathbb{R}_{\geq 0}$ and $T \in \mathbb{R}_{>1}$ for all $t \in \mathbb{R}_{\geq 0}$, which makes the following inequality hold*

$$|\Xi_a(0, t)| \leq T_0 + \frac{t}{T}. \tag{4}$$

**Assumption 2** (*DoS Frequency* [22]). *There exist $m(t)$, $T_{D_0} \in \mathbb{R}_{\geq 0}$ and $T_D \in \mathbb{R}_{>0}$ for all $t \in \mathbb{R}_{\geq 0}$, which makes the following inequality hold*

$$m(0, t) \leq T_{D_0} + \frac{t}{T_D}. \tag{5}$$

**Remark 1.** *The intent of the DoS attack is not generally sporadic and periodic, but aperiodic (stochastic). Therefore, the periodic DoS attack is not realistic in theoretical research and practical industrial production. To tackle this issue, this paper focuses on the more realistic aperiodic DoS attacks in the following part. In addition, Assumption 1 excludes the situation of continuous DOS attacks, which makes the considered system open-loop and uncontrollable.*

In this subsection, we will illustrate the proposed procedures for a resilient dynamically triggering strategy in favor of the aperiodic DoS attacks. To achieve this goal, denote $e$ the difference between the last successfully transmission state $x(t_i)$ and the current state $x(t)$

$$e = x(t_i) - x(t), \forall t \in [t_i, t_{i+1}), i \in \mathbb{Z}, t_0 = 0 \tag{6}$$

where $t_i$ is determined by the upcoming DTM (7).

$$t_{i+1} = \inf\{t > t_i, t \in R | g(e) \leq 0\}. \tag{7}$$

Herein, similar to [9], using the trigger function $g(e) = \eta + \theta(\rho(1 - c_1)\alpha(x) - \gamma(\|e\|)), \theta \in \mathbb{R}^+$. Additionally, the internal dynamic variable $\eta$ is to be defined before a new dynamic triggering strategy sprung up, which is

$$\dot{\eta} = -\lambda\eta + \rho(1 - c_1)\alpha(\|x\|) - \gamma(\|e(t)\|), \ \eta(0) = 0 \tag{8}$$

where $\eta$ is a locally Lipschitz continuous $\mathcal{K}_\infty$ function. Intuitively, $\eta$ may be regarded as a filtered value of $\alpha(1 - c_1)\alpha(\|x\|) - \gamma(\|e(t)\|)$ (refers to [5]). In particular, the filter (8) is possibly nonlinear if the $\eta$ is nonlinear. The dynamic event-triggered strategy (7) reduces to event-triggered strategy in [29] when $\theta$ goes to $+\infty$ (detailed analysis refers to [9]).

Based on the discussion before, the following lemma is needed to guarantee $\eta \geq 0$.

**Lemma 1.** *The variable $\eta$ defined in* (8) *is always non-negative.*

**Proof.** According to (8), which corresponds to the following inequality:

$$\eta + \theta(\rho(1 - c_1)\alpha(\|x\|) - \gamma(\|e(t)\|)) \geq 0. \tag{9}$$

First, if $\theta = 0$, then $\eta \geq 0$ is true.
Second, if $\theta \neq 0$, by combing (8) and (9), one has

$$\dot{\eta} + \lambda\eta = \rho(1 - c_1)\alpha(\|x\|) - \gamma(\|e(t)\|) \geq -\frac{\eta}{\theta}, \ \eta(0) \geq 0. \tag{10}$$

Then, solve (10) for $t \in [0, +\infty)$, one has

$$\eta(t) \geq \eta(0)e^{-(\lambda + \frac{1}{\theta})t}$$

which means that $\eta$ is lower bound by a positive exponential signal, so one can obtain $\eta \geq 0$. $\square$

In addition, taking DoS attacks into consideration, as we all know, the measurement data will be lost even if the condition (7) is satisfied. To alleviate the effects of DoS attacks, a resilient strategy will be presented in the following. In particular, combining with (7), one defines a novel RDET communication strategy as follows:

$$t_{i+1} = \{t_{i+1} \text{ satisfies } g(e) \leq 0 | t_{i+1} \in \mathcal{H}_n\} \cup \{t_m\}. \tag{11}$$

**Remark 2.** *With the opening of network control system communication, the system is more vulnerable to all kinds of malicious attacks. In order to eliminate the adverse effects of the attack and ensure better performance of the system, the elastic control technology based on dynamic event triggering plays an important role. This is especially true in many industrial controls, such as power systems [27], Chua circuits [30], and vehicle systems [31].*

Next, we use the following DoS attacks as follows:

$$\mathcal{W}(t) = \begin{cases} 0, & t \in [t_m, t_m + \Delta_m) \\ 1, & t \in [t_m + \Delta_m, t_{m+1}). \end{cases} \tag{12}$$

In this paper, the state-dependent control input $u(t) = k(x(t))$ under DoS attacks can be represented as

$$u(t) = (1 - \mathcal{W}(t))k(x(t)). \tag{13}$$

Based on the above analysis, in what follows, combining (1), (12), and (13), the switched version of system (1) can be represented as

$$\dot{x}(t) = f(x(t), (1 - \mathcal{W}(t))k(x(t))), \tag{14}$$

In the following sections, the conditions for the exponential stability of system (14) with DoS attacks will be provided.

### 3. Main Result

This section aims to develop a piecewise Lyapunov function for NNS under DoS attacks. The resilient analysis of nonlinear switched system (14) is discussed, and the related parameters are obtained. Theorem 1 is presented to guarantee the RDET control for NNS under DoS attacks. In addition, it is worth noting that the Zeno behavior is excluded in Theorem 2.

**Theorem 1.** *Consider the NNSs* (1) *under DoS attacks satisfying Assumptions 1 and 2, under the switched controller* (13) *with dynamic event-triggered condition* (7). *If some $\mathcal{K}_\infty$ functions $\alpha_1$, $\alpha_2$ and $\gamma$ hold, then the switched Lyapunov function $V_j$, $j = 1, 2$ of* (14) *satisfies the following inequality*

$$\alpha_1(\|x\|) \leq V_j(x) \leq \alpha_2(\|x\|) \tag{15}$$

$$\nabla V_1(x) \cdot f(x, k(x + e)) \leq -\omega_1 V(x) + \gamma(\|e\|), \tag{16}$$
$$t \in \Xi_s(0, t)$$

$$\nabla V_2(x) \cdot f(x, 0) \leq \omega_2 V_2(x), t \in \Xi_a(0, t) \tag{17}$$

$$V_1(x) \leq \mu_1 V_2(x^-), V_2(x) \leq \mu_2 V_1(x^-) \tag{18}$$

*where $\mu_1, \mu_2 \geq 1, \omega_1 > 0, \omega_2 > 0$ and the parameters $T_D$ in* (6) *and $T$ in* (4) *satisfy*

$$\frac{\ln(\mu_1 \mu_2)}{T_D} + \frac{(\omega_1 + \omega_2)}{T} \leq \omega_1. \tag{19}$$

*Then, system* (14) *is GWES. In particular*

$$\|x(t)\| \leq \alpha_1^{-1}\left(Me^{-\beta_1 t}\alpha_2(\|x(0)\|)\right). \tag{20}$$

**Proof.** To show the complete theoretical analysis of the above theorem, we will deal with it in two steps.

Case 1: Assume there are no DoS attacks.

For $t \in \mathcal{H}_m$, according to (16) and DTM (11), the derivative of $V_1(x(t))$ is subject to

$$\begin{aligned}
\dot{V}_1(x(t)) &\leq -c_1 V_1(x(t)) + \gamma(\|e\|) + \dot{\eta} \\
&\leq -\rho c_1 V_1(x(t)) - c_1(1 - \rho)V_1(x(t)) + \gamma(\|e\|) \\
&\quad - \lambda \eta + c_1(1 - \rho)V_1(x(t)) - \gamma(\|e\|) \\
&\leq -\rho c_1 V_1(x(t)) - \lambda \eta \\
&\leq -\omega_1 V_1(x(t)) \tag{21}
\end{aligned}$$

where $\rho \in (0, 1)$ and $\omega_1 = c_1 \rho$.

Case 2: Assume there are DoS attacks.

For $t \in \mathcal{D}_m$, based on (17), it is easy to obtain the derivative of $V_2(x(t))$ as follows

$$\dot{V}_2(x(t)) \leq \omega_2 V_2(x(t)). \tag{22}$$

Hence, combing (21) with (22) gives a piecewise Lyapunov functional, $t \in \mathcal{H}_m$ and $t \in \mathcal{D}_m$, respectively, can be found as below

$$V(x(t)) = \begin{cases} e^{-\omega_1(t-t_m)}V(x(t_m)), t \in \mathcal{H}_m \\ e^{\omega_2(t-t_{m-1}-\Delta_{m-1})}V(x(t_{m-1}+\Delta_{m-1})), \\ \qquad\qquad\qquad\qquad t \in \mathcal{D}_m. \end{cases}$$

First, assume $t \in \Xi_s(0,t)$, according to definition 4, one has

$$\begin{aligned} V_1(x(t)) &\le e^{-\omega_1(t-t_m)}V_1(x(t_m)) \\ &\le \mu_1 e^{-\omega_1(t-t_m)}V_2(x^-(t_m)) \\ &\le \mu_1 e^{-\omega_1(t-t_m)}e^{\omega_2(t_m-t_{m-1}-\Delta_{m-1})} \\ &\quad V_2(x(t_{m-1}+\Delta_{m-1})) \\ &\le \mu_1\mu_2 e^{-\omega_1(t-t_m)}e^{\omega_2(t_m-t_{m-1}-\Delta_{m-1})} \\ &\quad V_1(x^-(t_{m-1}+\Delta_{m-1})) \\ &\qquad\qquad \vdots \\ &\le (\mu_1\mu_2)^{m(0,t)}e^{-\omega_1(t-t_m+\Delta_{m-1}+\cdots+\Delta_0)} \\ &\quad e^{\omega_2(t_m-\Delta_{m-1}-\cdots-\Delta_0)}V(x(0)) \\ &\le (\mu_1\mu_2)^{T_{D_0}+\frac{t}{T_D}}e^{(\omega_1+\omega_2)T_0} \\ &\quad e^{(\omega_1+\omega_2)\frac{t}{T}-\omega_1 t}V(x(0)) \\ &= M_1 e^{[\frac{\ln(\mu_1\mu_2)}{T_D}+\frac{(\omega_1+\omega_2)}{T}-\omega_1]t}V(x(0)) \end{aligned}\qquad(23)$$

where $M_1 = (\mu_1\mu_2)^{T_{D_0}}e^{(\omega_1+\omega_2)T_0}$.

Then, assume $t \in \Xi_a(0,t)$, according to definition 4 again, one has

$$\begin{aligned} V_2(x(t)) &\le e^{\omega_2(t-t_{m-1}-\Delta_{m-1})}V_2(x(t_{m-1}+\Delta_{m-1})) \\ &\le \mu_2 e^{\omega_2(t-t_{m-1}-\Delta_{m-1})}V_1(x^-(t_{m-1}+\Delta_{m-1})) \\ &\le \mu_2 e^{\omega_2(t-t_{m-1}-\Delta_{m-1})}e^{-\omega_1\Delta_{m-1}}V_1(x(t_{m-1})) \\ &\le \mu_1\mu_2 e^{\omega_2(t-t_{m-1}-\Delta_{m-1})}e^{-\omega_1\Delta_{m-1}}V_2(x^-(t_{m-1})) \\ &\qquad\qquad \vdots \\ &\le \mu_1^{m(0,t)-1}\mu_2^{m(0,t)}e^{-\omega_1(\Delta_{m-1}+\cdots+\Delta_0)} \\ &\quad e^{-\omega_2(t-\Delta_{m-1}-\cdots-\Delta_0)}V(x(0)) \\ &\le \mu_1^{m(0,t)-1}\mu_2^{m(0,t)}e^{-\omega_1(t-\Xi_a)}e^{-\omega_2\Xi_a}V(x(0)) \\ &= M_2 e^{[\frac{\ln(\mu_1\mu_2)}{T_D}+\frac{(\omega_1+\omega_2)}{T}-\omega_1]t}V(x(0)) \end{aligned}\qquad(24)$$

where $M_2 = \mu_1^{-1}(\mu_1\mu_2)^{T_{D_0}}e^{(\omega_1+\omega_2)T_0}$.

Finally, combining (23) and (24), one has

$$V(x(t)) \le M e^{-[\omega_1-\frac{\ln(\mu_1\mu_2)}{T_D}-\frac{(\omega_1+\omega_2)}{T}]t}V(x(0))\qquad(25)$$

where $M = \max\{M_1, M_2\}$.

In what follows, using (15), the above inequality (25) can be modified as

$$\alpha_1(\|x(t)\|) \leq V(x(t)) \leq Me^{-\beta_1 t}V(x(0))$$
$$\leq Me^{-\beta_1 t}\alpha_2(\|x(0)\|) \tag{26}$$

where $\beta_1 = \omega_1 - \frac{\ln(\mu_1\mu_2)}{T_D} - \frac{(\omega_1+\omega_2)}{T}$. Based on (26), if DoS attacks satisfy $\frac{\ln(\mu_1\mu_2)}{T_D} + \frac{(\omega_1+\omega_2)}{T} \leq \omega_1$, then, system (1) with control input (11) under DoS attacks is GWES. Further, one has

$$\|x(t)\| \leq \alpha_1^{-1}(Me^{-\beta_1 t}\alpha_2(\|x(0)\|)).$$

The proof is completed. □

**Remark 3.** *This theorem characterizes the system's resilience issue. Moreover, $\frac{\ln(\mu_1\mu_2)}{T_D} + \frac{(\omega_1+\omega_2)}{T} \leq \omega_1$ shows that the stability of NNSs can be guaranteed in the event they suffer from more DoS attacks that satisfy some constraints of attack interval and attack frequency.*

**Remark 4.** *It is worth pointing out that the inequality constraint on $\alpha_1$ and $\gamma$ in Assumption 4 in [23] is unnecessary in our work. More specifically, in this article, with the aid of introducing a piecewise Lyapunov function, Assumption 4 in [23] is removed, which reduces the conservatism of the system.*

Next, we will give the conditions that void the Zeno behavior. Before continuing the discussion, we impose an assumption.

**Assumption 3** ([5]). *Because of the Lipschitz continuity of the function $f(x,u)$, there exists a constant $L_1$ which satisfies the following inequality*

$$\|f(x,u\| = \|f(x+k(x+e_i))\|$$
$$\leq L_1(\|x\| + \|e_i\|) \tag{27}$$

*where $u(t) = k(x(t_i)), t \in [t_i, t_{i+1})$.*

**Theorem 2.** *For the NNSs (1) under the event-triggered strategy (11) and the controller (13), there exists a minimal inter-event time $\tau$ ensures that Zeno behavior does not exist, where $\tau$ is given by*

$$\tau \geq \frac{1}{2L_1}\ln(\frac{(2\gamma^{-1}(\eta/\theta + \rho(1-c_1)\alpha(x))}{\|x(t_i)\|} + 1).$$

**Proof.** First, we define the inter-execution time $\tau = t_{i+1} - t_i$. According to the ZOH scheme, there is $\dot{e}_i = 0$ when $t = t_i$. Meanwhile, with the inequality (27), one has

$$\|\dot{x}\| = \|f(x,u(x+e_i))\|$$
$$\leq L_1(\|x\| + \|e_i\|). \tag{28}$$

Furthermore, for $\forall t \in [t_i, t_{i+1})$, it is easy to get the following equation

$$\dot{x} = -\dot{e}_i. \tag{29}$$

Next, combining (6), (28), and (29), one has

$$\|\dot{e}_i\| \leq L_1(\|x(t_i) - e_i\| + \|e_i\|).$$

Herein, using the comparison lemma, we have

$$\|e_i\| \leq \frac{\|x(t_i)\|(e^{2L_1(t-t_i)} - 1)}{2}. \tag{30}$$

With (11) associated with

$$\eta + \theta(\rho(1 - c_1)\alpha(x) - \gamma(\|e_i\|)) \leq 0,$$

from this inequality (31), we get

$$\|e_i\| \geq \gamma^{-1}(\eta/\theta + \rho(1 - c_1)\alpha(x)). \tag{31}$$

Finally, combining with (30), (31), and $t - t_i \leq \tau, t \in [t_i, \ t_{i+1})$, one has

$$\tau \geq \frac{1}{2L_1}\ln\left(\frac{(2\gamma^{-1}(\eta/\theta + \rho(1 - c_1)\alpha(x))}{\|x(t_i)\|} + 1\right),$$

where $\eta > 0$, $\theta > 0$. The proof is completed. $\square$

## 4. Simulation

In this section, the practical merits of the proposed controllers and theory are corroborated using the well-known nonlinear Chua circuit, as shown in Figure 3. Considering the control input $u = (u_1; u_2; u_3)$, its dynamics are generated as

$$\begin{cases} \dot{v}_1 = \dfrac{v_2 - v_1}{RC_1} - \dfrac{f(v_1)}{C_1} + u_1, \\[2mm] \dot{v}_2 = \dfrac{v_1 - v_2}{RC_2} + \dfrac{v_3}{C_2} + u_2, \\[2mm] \dot{i}_3 = -\dfrac{v_2}{L} - \dfrac{R_0 v_3}{L} + u_3 \end{cases} \tag{32}$$

where $v_1$ and $v_2$ are voltages across $C_1$ and $C_2$, respectively. $i_3$ is current through the inductance. $f(v_1) = g_1 v_1 + g_2 v_1^3$ is characteristic of the nonlinear resistance $R_N$.

Next, let $x_1 = v_1, x_2 = v_2, x_3 = i_3$, and $x = (x_1; x_2; x_3)$. Then, (32) can be transferred into the following as

$$\begin{cases} \dot{x}_1 = \dfrac{x_2 - x_1}{RC_1} - \dfrac{g_1 x_1 + g_2 x_1^3}{C_1} + u_1, \\[2mm] \dot{x}_2 = \dfrac{x_1 - x_2}{RC_2} + \dfrac{x_3}{C_2} + u_2, \\[2mm] \dot{x}_3 = -\dfrac{x_2}{L} - \dfrac{R_0 x_3}{L} + u_3 \end{cases} \tag{33}$$

According to [33], there exists a chaotic attractor (see Figure 4) when some parameters are fixed at $C_1 = 0.7; C_2 = 7.8; L = 1.897; R_0 = 0.01499; g_1 = -0.59; g_2 = 0.02; R = 2.1;$ $u = 0$. Meanwhile, system (33) is rewritten as

$$\begin{cases} \dot{x}_1 = 0.1626x_1 + 0.6803x_2 - 0.0286x_1^3 + u_1, \\ \dot{x}_2 = 0.0611x_1 - 0.0611x_2 + 0.1282x_3 + u_2, \\ \dot{x}_3 = -0.5271x_2 - 0.079x_3 + u_3. \end{cases} \tag{34}$$

Set the initial state $x_0 = (-0.6061; -0.3483; 0.6013)$. System (34) is unstable without the control input $u$. Then, set $\theta = 0.1; \eta = 0.1; \lambda = 0.5; \rho = 0.01$; simulation time $[0, 200s]$ with sampling period $h = 0.05s$.

Case 1: According to Theorem 1, one can design a controller $u = [-0.01x_1; 0; -0.1x_3]$ under the RDET strategy (11) to stabilize system (34) without aperiodic DoS attacks. The

state response of system (34) is shown in Figure 5. Moreover, the number of triggered packets to be transmitted is 82 times. The event interval time of the event generator is depicted in Figure 6. In particular, one can design another controller $u = [-0.1x_1; 0; -0.1x_3]$ under event-triggered communication scheme (7) to stabilize system (34) without DoS attacks. Meanwhile, a stable periodic solution will be presented in Figure 7. The event intervals of the event generator are depicted in Figure 8.

Case 2: In the sequence, under the same circumstances, once system (34) suffers from malicious aperiodic attacks, the system is unstable in Figure 9, where the gray areas represent the DoS attack time intervals. In addition, release time intervals are depicted in Figure 10 with 756 triggered packets to be successfully transmitted to the controller under the event-triggered communication scheme (7).

Case 3: The system is not GAS with $u = 0$. The stabilizing control law is $u = [-2x_1; -2x_2; \ -x_3]$. We select $V_1(x) = \frac{1}{2}\|x\|^2$ as a Lyapunov function when $t \in \mathcal{H}_m$, so that the $V_1(x) > 0$ holds true if $\|x\| \neq 0$. Notice that

$$
\begin{aligned}
\nabla V_1(x)f(x,u) =\ & x_1 \cdot \dot{x}_1 + x_2 \cdot \dot{x}_2 + x_3 \cdot \dot{x}_3 \\
=\ & -1.8374x_1^2 + 0.7414x_1x_2 - 2.0611x_2^2 \\
& -1.079x_3^2 - 0.3989x_2x_3 - 0.0286x_1^4 \\
& -2x_1e_1 - 2x_2e_2 - x_3e_3 \\
\leq\ & -0.4667x_1^2 - 0.4909x_2^2 - 0.8795x_3^2 \\
& + e_1^2 + e_2^2 + e_3^2 \\
\leq\ & -0.403V_1(x) + \|e\|^2.
\end{aligned}
\tag{35}
$$

Next, We select $V_2(x) = \frac{1}{2}(x_1^2 + x_2^2 + 2x_3^2)$ as Lyapunov function when $t \in \mathcal{D}_m$, so that the $V_2(x) > 0$ holds true if $\|x\| \neq 0$. Notice that

$$
\begin{aligned}
\nabla V_2(x)f(x,0) =\ & x_1 \cdot \dot{x}_1 + x_2 \cdot \dot{x}_2 + 2x_3 \cdot \dot{x}_3 \\
\leq\ & 0.5333x_1^2 + 0.7726x_2^2 + 0.3050x_3^2 \\
\leq\ & 0.0314V_2(x).
\end{aligned}
\tag{36}
$$

According to the aforementioned analysis, there exist $\alpha_1(\cdot) = x_1^2 + x_2^2 + x_3^2$ and $\alpha_2(\cdot) = 2x_1^2 + 2x_2^2 + 2x_3^2$, which satisfy (15) in Theorem 1. Meanwhile, (35) and (36) allow $\omega_1 = 0.403$, $\omega_2 = 0.0314$ to satisfy (16) and (17) in Theorem 1, respectively. In addition, set $\beta_1 = 0.1165$, $\mu_1 = 1$ and $\mu_2 = 2$, which satisfy (18) in Theorem 1. Let $T_0 = 0.1$, $T = 2$, $T_{D_0} = 0.1$, and $T_D = 10$, one has $\frac{\ln 2}{10} + \frac{(0.403 + 0.0314)}{2} = 0.2865 < 0.403$ and $\|x(t)\| \leq \alpha_1^{-1}(Me^{-\beta_1 t}\alpha_2(\|x(0)\|))$ satisfy (19) and (20), respectively. Moreover, $m \leq 0.1 + \frac{200}{10} = 20.1$ and $\|x(t)\| \leq 0.1204e^{-0.233t}$.

Herein, Figure 11 depicts the state responses of system (34) under DoS attacks, and which shows that system (34) is stable. The release instants are depicted in Figure 12, and there are 716 sampled packets transmitted successfully Under the supervision of the DTM (11). Figure 13 presents the aperiodic DoS attack sequence. Next, Table 1 presents a comparison of the different triggering strategies.

First, before analyzing Table 1, we give out design formulas on a triggering rate, which is expressed as $\frac{\text{event}_{\text{number}}}{\text{sample}_{\text{number}}}$. Second, from Table 1, it is obvious that once the system suffers from aperiodic DoS attacks, the number of triggering events and the triggering rate will add the same parameters designed before to the framework. On contrary, the average interval will become small to add the number of triggering events and compensate for lost packets due to DoS attacks.

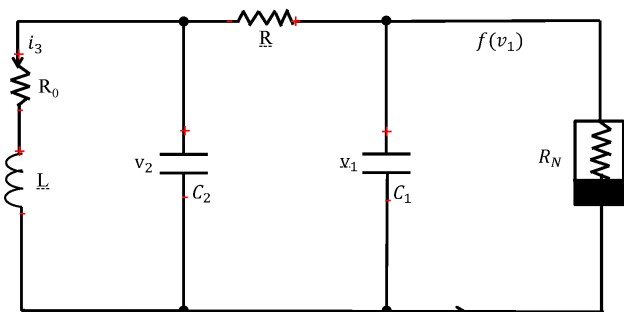

**Figure 3.** Diagram of a nonlinear Chua circuit.

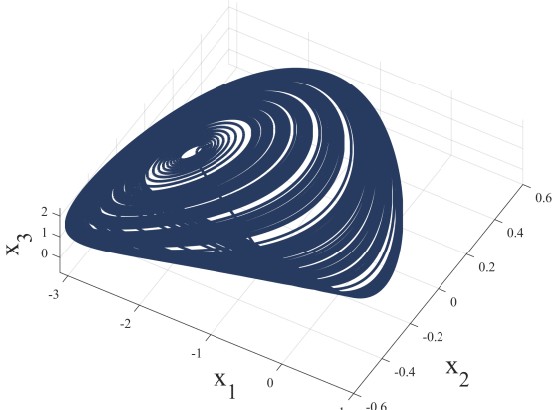

**Figure 4.** The chaotic attractor for the Chua circuit.

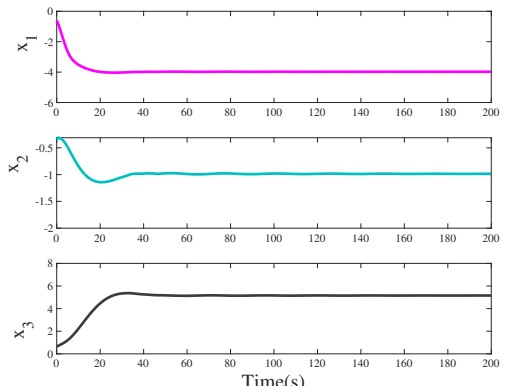

**Figure 5.** State responses of triggering control systems without aperiodic DoS attacks.

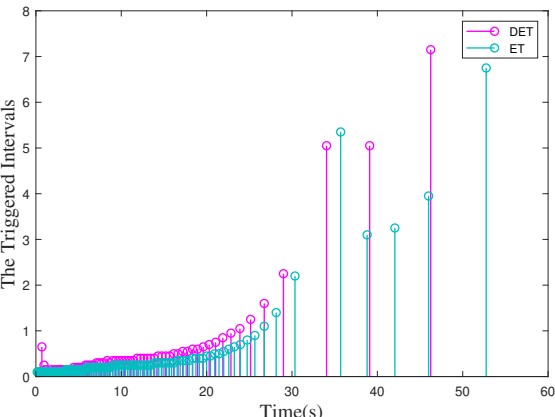

**Figure 6.** The triggered intervals of control systems without aperiodic DoS attacks.

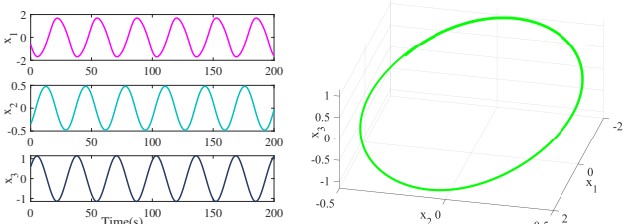

**Figure 7.** Stable periodic solution of triggering control systems without aperiodic DoS attacks.

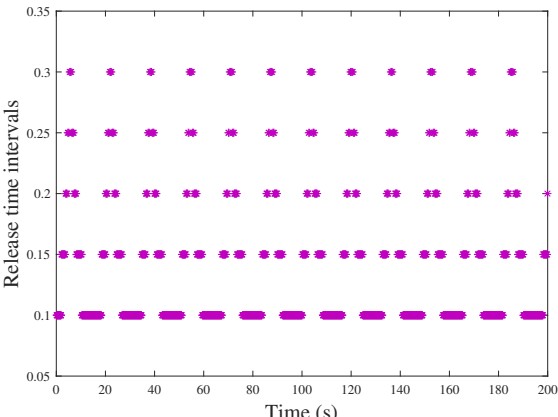

**Figure 8.** Release time intervals corresponding to a stable periodic solution.

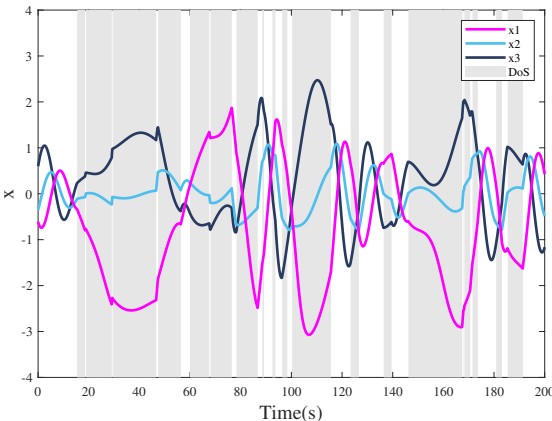

**Figure 9.** The state trajectories of unstable triggering control systems under aperiodic DoS attacks.

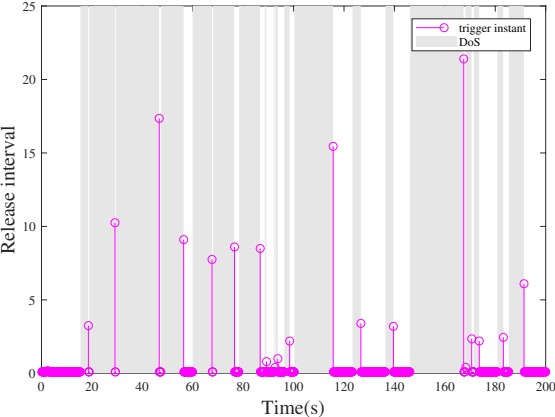

**Figure 10.** Period of triggering control systems under aperiodic DoS attacks.

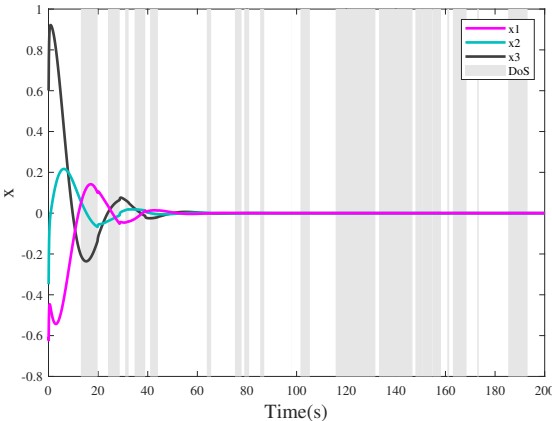

**Figure 11.** The stable state trajectories under aperiodic DoS attacks.

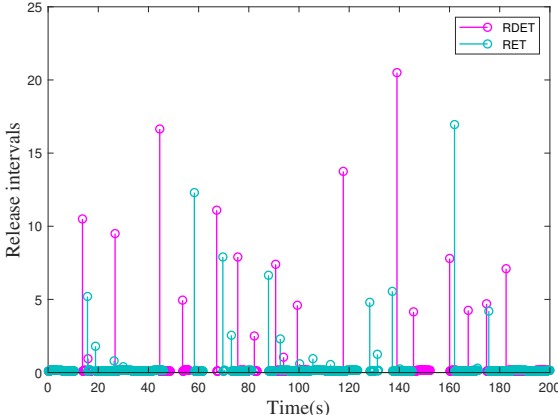

**Figure 12.** Triggered instants and release intervals under aperiodic DoS attacks.

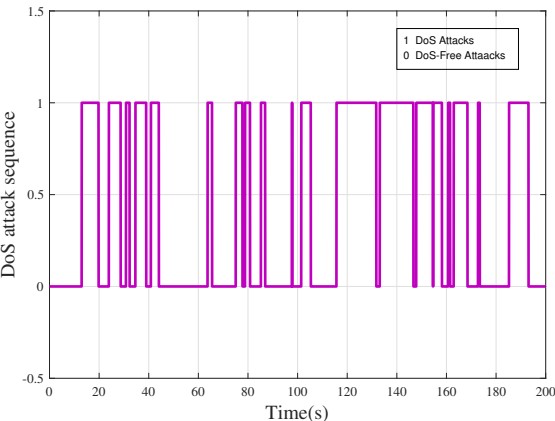

**Figure 13.** Aperiodic DoS attacks.

**Remark 5.** *In the simulation, the modeling and generation of DoS attacks is similar to [22]. Since the attacks are affected by energy constraints, they are intermittent non-periodic attacks, and only focus on the single communication channels (sensor-to-controller).*

**Remark 6.** *Figures 6 and 12 display the triggered intervals of the four strategies. Figure 6 indicates RDET strategy can generate a bigger average interval than the ET strategy without aperiodic DoS attacks. Moreover, a similar result is presented in Figure 12, namely, the RDET strategy can generate a bigger average interval than the RET strategy suffering aperiodic DoS attacks. This result is consistent with Table 1.*

**Table 1.** Comparing different control schemes.

| Strategies | Trigger | Average Interval | Triggering Rate |
|---|---|---|---|
| ET in [5] | 120 | 1.67 | 3% |
| DET in [9] | 82 | 2.44 | 2.05% |
| RET in [22] | 960 | 0.21 | 24% |
| DTRC in this work | 716 | 0.28 | 17.9% |

## 5. Conclusions

In this paper, we have designed a DRTC to stabilize NNSs under malicious aperiodic DoS attacks. Furthermore, the stability criterion is obtained under malicious aperiodic DoS attacks based on Lyapunov theory. In addition, the minimal inter-event time $\tau$ excludes Zeno behavior for the controller (13) with dynamically triggering strategy (11). Finally, the merits of the proposed controllers and theory are corroborated using the well-known nonlinear Chua circuit. Based on our current work, in the future, we will consider security-based event-triggered learning control for NNSs subject to stochastic attacks.

**Author Contributions:** Conceptualization, W.T.; methodology, W.T.; validation, W.T., H.W. and H.H.; formal analysis, H.W., H.H. and X.L.; investigation, W.T. and H.H.; writing—original draft preparation, W.T.; writing—review and editing, W.T., H.W. and H.H.; visualization, W.T., H.W., X.L. and M.Z.; supervision, H.W., H.H. and X.L.; project administration, H.W. and H.H.; funding acquisition, H.W., H.H., X.L. and M.Z. All authors have read and agreed to the published version of the manuscript.

**Funding:** This work was supported by the National Natural Science Foundation of China under Grant Nos. 61673107, 62073076, 62203109, 62003218, and 62022088, International Partnership Program of Chinese Academy of Sciences under Grant 173321KYSB20200002, the Jiangsu Provincial Key Laboratory of Networked Collective Intelligence under Grant No. BM2017002, Guangdong Basic, Applied Basic Research Foundation under Grants 2019A1515110234 and 2020A1515110148, and Natural Science Foundation of Jiangsu Province under grant numbers BK20210216 and BK20220812.

**Institutional Review Board Statement:** Not applicable.

**Informed Consent Statement:** Not applicable.

**Data Availability Statement:** Not applicable.

**Conflicts of Interest:** The authors declare no conflict of interest.

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
