# Peer review of "Dynamically Triggering Resilient Control for Networked Nonlinear Systems under Malicious Aperiodic DoS Attacks"

_processes, doi:10.3390/pr10122627_

Round 1
Reviewer 1 Report
- The abstract of a paper should highlight the major need for performing the solution over DoS problem. The author should concise it properly and avoid the use of "some" word.
-The introduction of a paper should summarize all the previous techniques used in the discussed field and should contain a major portion of references. However, only a few references have been cited in the Introduction of the paper.
- The paper should emphasis on critical analysis of the existing work with its merits and demerits.
-The conclusion section is not properly written and also future plans with respect to the research state of progress
Reviewer 2 Report
To improve the quality of the paper, the reviewer gives the following comments.
(1) In Section 1, the paper claims “the proposed method not only releases the assumption 4 in [28] but also reduces the conservative of the system”. The advantage and disadvantage of reducing the conservative of the system should be discussed in detail.
(2) More application scenarios of the proposed resilient dynamic event-triggered communication strategy need to be introduced.
(3) The paper should explain the reasonableness of Assumption 1.
(4) In Section 3, it has “In addition, it is worthing noting, Zeno behavior is excluded in Theorem 2”. However, Theorem 2 is not found in this paper.
(5) The efficiency of the proposed controller is not analyzed and compared with other similar controllers. It also needs some generally accepted performance indexes to evaluate these controllers.
(6) In Section 4, the paper should present how to generate the DoS attacks in the simulation.
(7) The conclusion section should also give some hints for future work.
(8) Authors should proofread the paper against typos and grammar errors. The examples of this problem include “Finally, Some simulation” in Abstract, “control for nonlinear systems under DoS attacks [28? –30]” and “[? ] investigates the fuzzy resilient control” in Page 2, and “The following definitions and assumptions are provided for later analysis. ([? ] )” in Page 3. It would be good if a fluent English-writer could go over the paper to correct errors.
Round 2
Reviewer 2 Report
English language still needs to be polished. The writing errors exist in the revision version, for example, “R+ and Z+ represents the sets of the positive real numbers” in line 83 and “Then, an aperiodic DoS attacks scenarios is typically” in lines 91-92.
